# Irrigation of biomass plantations may globally increase water stress more than climate change

Fabian Stenzel [1,2,3,4 ✉], Peter Greve [2], Wolfgang Lucht [1,3,4], Sylvia Tramberend [2], Yoshihide Wada [2] & Dieter Gerten [1,3,4]

Bioenergy with carbon capture and storage (BECCS) is considered an important negative emissions (NEs) technology, but might involve substantial irrigation on biomass plantations. Potential water stress resulting from the additional withdrawals warrants evaluation against the avoided climate change impact. Here we quantitatively assess potential side effects of BECCS with respect to water stress by disentangling the associated drivers (irrigated biomass plantations, climate, land use patterns) using comprehensive global model simulations. By considering a widespread use of irrigated biomass plantations, global warming by the end of the 21st century could be limited to 1.5 °C compared to a climate change scenario with 3 °C. However, our results suggest that both the global area and population living under severe water stress in the BECCS scenario would double compared to today and even exceed the impact of climate change. Such side effects of achieving substantial NEs would come as an extra pressure in an already water-stressed world and could only be avoided if sustainable water management were implemented globally.

[1] Potsdam Institute for Climate Impact Research (PIK), Member of the Leibniz Association, Potsdam, Germany. [2] International Institute for Applied Systems Analysis (IIASA), Laxenburg, Austria. [3] Humboldt-Universität zu Berlin, Department of Geography, Berlin, Germany. [4] Integrative Research Institute on Transformations of Human-Environment Systems, Berlin, Germany. ✉email: stenzel@pik-potsdam.de

The Earth system is facing multiple environmental pressures (e.g. climate change, water shortages, ecosystem degradation), while the need remains to ensure food and water security for a growing world population. Additionally, there is growing interest in NE technologies linked to the desire to achieve the 1.5 °C target without jeopardizing sustainable development goals (SDGs) such as achieving water security. These challenges and their prospective solutions are intrinsically coupled, requiring strong trade-offs to be resolved. One of these dilemmas is centered around freshwater availability and stress. Water stress—affecting about 1.4–4 bn people already depending on the chosen metric[1–4]—may strongly increase in the future not only due to population growth, but also due to impacts of global climate change[5–8]. For example, a further 8% of world population may be exposed to increasing water stress due to climate change alone[9]. While mitigation of climate change will thus be imperative to reduce the pressure on freshwater resources (among other benefits)[10], the currently pledged emission reductions may not be enough to limit mean global warming to below 2 °C as envisaged in the Paris Agreement[11,12], requiring further measures such as active plant-based $CO_2$ sequestration from the atmosphere through dedicated biomass plantations combined with carbon capture and storage (BECCS)[13–15]. BECCS is based on the cultivation of fast-growing plant species, which are assumed to be regularly harvested for their biomass and subsequently processed to bio-fuels (replacing liquid fossil energy carriers), or burned for energy generation (offsetting coal or gas power plants), while the released $CO_2$ is (at least partially) captured[16,17]. Thus the whole process would remove $CO_2$ from the atmosphere and counteract anthropogenic green-house gas emissions to reduce climate change. The sequestration potential was estimated to be 0.1 GtC yr$^{-1}$ to 2 GtC yr$^{-1}$ by 2050 and 0.3 GtC yr$^{-1}$ to 3.3 GtC yr$^{-1}$ by 2100[18,19]. Utilization of biomass is supposed to provide substantial amounts of electric energy or liquid fuels (up to 500 EJ yr$^{-1}$), and is thus assumed to be deployed at large-scale (even without providing negative emissions via CCS) and also rather early in the 21st century (together with afforestation) in contrast to more expensive NE technologies like direct air capture[20].

However, at the large-scale required, biomass production is likely to increase the pressure along multiple environmental dimensions locally and globally[21–23], including increased competition for scarce freshwater resources to the extent that such plantations require irrigation in order to reach anticipated sequestration levels[24,25]. From a sustainability perspective, it is important to understand how additional water use for bioenergy production affects water stress in relation to the avoided change that would occur in a warming world without irrigated biomass plantations.

We define water stress using an established globally applicable metric: the local ratio of total human water withdrawals to available discharge[3,26,27], from which the yearly mean water stress is derived.

To corroborate findings from one earlier regional study that suggested the water stress in a mitigation scenario based on irrigated bioenergy may indeed supersede that of unabated climate change[28], we here provide a systematic global-scale analysis comparing water stress in two plausible future scenarios: a world with strong mitigation including (partially irrigated) bioenergy plantations (~600 Mha in 2095) as a contribution to limit mean global warming by the end of the century to around 1.5 °C (hereinafter referred to as scenario *BECCS*), and one with only marginal extent of bioenergy plantations (~30 Mha in 2095) resulting in warming of 3 °C (*CC*).

We thus advance earlier studies[28–30] by globally and spatially explicitly comparing water stress and its drivers between a strong climate change scenario with one where bioenergy is used for mitigation. We take into account available surface water restrictions (e.g. to safeguard environmental flow requirements of river ecosystems) for the irrigation of biomass plantations and cropland. This approach enables us to highlight and quantify trade-offs regarding different levels of water protection, impacts of climate change versus mitigation through BECCS, and also the possible contribution of improved water management to help solve this dilemma. Unlike previous BECCS water demand studies[31,32], we apply transient land use projections for both bioenergy and food crops[33], which are consistent with future pathways of green-house gas emissions and socio-economic development.

The scenarios are based on data from the Representative Concentration Pathways RCP2.6 (*BECCS*) and RCP6.0 (*CC*), both following the "middle of the road" narrative of the Shared Socioeconomic Pathway SSP2, provided by the Inter-Sectoral Impact Model Intercomparison Project (ISIMIP2b)[33]. Scenarios differ in terms of the degree of climate change, BECCS deployment, and land use change trajectories over the 21st century including differences in the spatial distribution of (rainfed and irrigated) areas with agricultural crops and biomass plantations (Table 1).

To study the beneficial effects of more sustainable water use policies while providing the same amount of biomass as in scenario *BECCS*, we additionally explore a scenario with irrigated bioenergy plantations that are accompanied by sustainable water management (*BECCS+SWM*), while all other parameters are chosen for maximum consistency with scenario *BECCS*. This scenario assumes the preservation of environmental flow requirements (EFRs) and implements advanced on-field water management[32,34] on both agricultural and bioenergy sites. EFRs determine a percentage of pristine, undisturbed mean monthly river flow, here following a *variable monthly flow* (VMF) method[35].

Fractions of the local biomass plantation area that are equipped for irrigation (30%—*BECCS*, 45%—*BECCS+SWM*) are obtained from a sensitivity analysis, assuming that 50% of the required harvest increase between the *Baseline* and the ISIMIP2b harvests including technological change is achieved by irrigation (for more details see Methods—determining the bioenergy irrigation amount).

The simulations are performed using the process-based global vegetation and water balance model LPJmL[36] forced by climate change scenarios from four General Circulation Models (GCMs) selected in the ISIMIP2b project: HadGEM2-ES, MIROC5, GFDL-ESM2M, IPSL-CM5A-LR. We use an ensemble of GCMs to account for the remaining variation in precipitation projections inherent to GCMs, even when forced with the same RCP[37,38].

We compute the water stress index (WSI) for each 0.5 × 0.5 degree grid cell as monthly averages of the present period 2006 to 2015 (*Today*) and the future period 2090 to 2099, expressed as percentages of human water use (withdrawals for

---

**Table 1 Scenario overview.**

| Scenario | CC | BECCS | BECCS +**SWM** |
|---|---|---|---|
| Climate forcing | RCP6.0 | RCP2.6 | RCP2.6 |
| Biomass plantation area (2090–2099) | 30 Mha | 600 Mha | 600 Mha |
| of which equipped for irrigation | 30% | 30% | 45% |
| Sustainable water management | No | No | Yes |

Climate, land use, and water use input data for 4 GCMs (all based on SSP2) is used from the ISIMIP2b project[33]. The irrigation fraction is obtained from a sensitivity analysis as part of this study. Sustainable water management is a combination of withdrawal restrictions based on EFRs[35], local water storage, and improved on-field irrigation efficiencies[32,34].

households, industry, and irrigation of biomass plantations and cropland) compared to total discharge. High stress is assumed to prevail in cells where the yearly mean WSI > 40%[26,39] (for more details see Methods – Water stress index WSI). From these cells, we calculate sums of global area as well as population under high water stress.

Here, we show that both the global area and the population exposed to high water stress would double in the *BECCS* scenario compared to today and even exceed the impact of climate change (scenario *CC*), unless sustainable water management was in place to reduce the pressure on freshwater resources.

## Results

**Globally aggregated results**. We find that by the end of this century (2090–2099), the global population and land area under high water stress will increase sharply in all scenarios without sustainable water management compared to the present (2006–2015). The total land area under high water stress—currently 1023 (982 to 1065) Mha—is simulated to increase in the inter-model mean to 1580 (1520 to 1613) Mha in *CC* and 1928 (1901 to 1970) Mha in *BECCS*. The number of people experiencing high water stress—currently 2.28 (2.23–2.32) billion people—increases to 4.15 (4.03–4.24) billion in *CC* and 4.58 (4.46–4.71) billion in *BECCS* (Fig. 1, Table S1).

Increases for population under high water stress include the effect of increased world population from 7 billion people in 2010 to 9 billion in 2100 according to SSP2[40].

**Global distribution of water stress**. In the following, we focus the presentation of results on simulations under HadGEM2-ES climate projections, which represents an intermediate model response to the applied emission scenario among the group of four GCMs (compare Supplementary Figs. 1 and 2). For results for all other GCMs we refer to the supplementary material (Supplementary Figs. 4, 5 and 6).

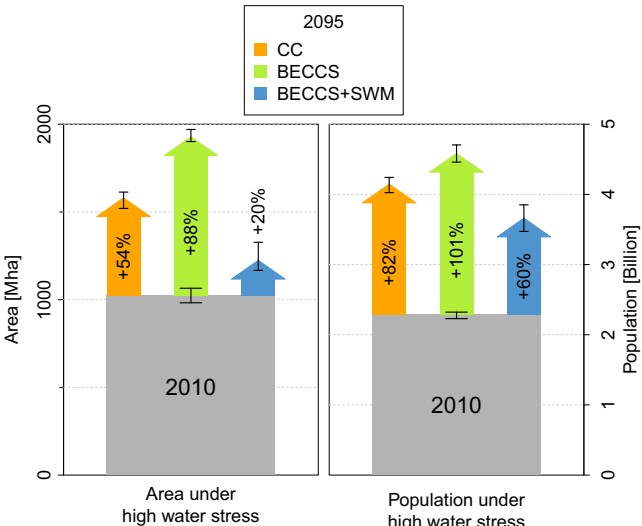

**Fig. 1 Simulated increase of area and population exposed to high water stress from around 2010 (2006–2015) to 2095 (2090–2099) in the different scenarios: CC (climate change), BECCS (bioenergy with carbon capture and storage), BECCS+SWM (BECCS with sustainable water management).** The numbers represent global sums of grid-cell-level area and population, respectively, where annual mean WSI > 40%. Shown are the mean change and the ranges resulting from the differences in climate simulations based on the four GCMs. Gray bars represent the current (2006–2015 average) levels.

The spatial distribution of locations with high water stress in the *CC* scenario is broadly similar to today's patterns, but the total area affected as well as the local WSI values increase significantly (Fig. 2), indicating that water stress in current hotspots will persist or even increase. Regional hotspots of WSI increases include the Mediterranean, the Middle East, India, North-East China, and South-East and southern West-Africa (Supplementary Fig. 7). In the *BECCS* scenario high water stress extends to otherwise unaffected regions (not highly stressed *Today* nor in *CC*) e.g. the East of Brazil and large parts of Sub-Saharan Africa (Fig. 2, Supplementary Fig. 8). These are regions where large-scale biomass plantations are assumed (according to the respective ISIMIP2b land use scenario for RCP2.6, see Methods—determining the bioenergy irrigation amount) and in which additional irrigation may therefore be required to increase biomass yields.

**Water stress differences between scenarios**. All future scenarios exhibit similar or higher water stress almost everywhere compared to *Today*, with only the Western United States and some locations in Asia showing the opposite behavior (Supplementary Figs. 7 and 8).

Globally, an area of about 2400 Mha (about 16% of the total land surface area) shows a difference larger than ±10% in WSI between the *BECCS* and *CC* scenarios. More than two-third (72%) of this area exhibits a higher WSI in the *BECCS* scenario (Fig. 3a), mostly located in Central and South America, Africa, and Northern Europe. Conversely, on less than one third (28%) of areas (Western US, India, South-East China, and a belt from the Mediterranean region to Kazakhstan), the *BECCS* scenario demonstrates lower water stress compared to the *CC* scenario, despite the irrigation for bioenergy.

Thus, without sustainable water management, irrigation of biomass plantations for the purpose of avoiding excessive climate change (3 °C vs. 1.5 °C) would increase water stress significantly in many regions (and also globally, Fig. 1). The effect of higher water stress due to irrigated biomass plantations is consistent among the different GCMs and ranges from 64% in IPSL, over 70% in GFDL, and 72% in HadGEM, to 79% in MIROC (Supplementary Fig. 1). These variations are potentially due to the precipitation and temperature differences between the GCMs (Supplementary Figs. 9 and 10).

**Drivers of water stress**. Higher WSI in *BECCS* compared to *CC* could result from differences in climate, land use, or the irrigation of biomass plantations, as these are the distinctive features in our experimental setup. To determine the attributing cause for the higher WSI in BECCS, we thus ran additional pairs of simulations only varying one of these features while fixing the others (see Methods—attribution of drivers for water stress differences and Fig. 4a–d). Globally, irrigated biomass plantations are the major driver for higher water stress in *BECCS* (see their extent in Fig. 5) due to the additional freshwater withdrawals. In regions which are simulated to experience a higher WSI in *CC*, differences are either due to land use or climate (with similar extent). Regarding the difference in land use patterns (Supplementary Fig. 11), we find a large increase in irrigation on areas of the food-producing agriculture (including pastures) in RCP2.6 vs. RCP6.0, which, for example, explains the patterns for the Western United States. The higher water stress in *CC* compared to *BECCS* due to climate differences (mostly in Asia) can be attributed to increases in water availability (see precipitation difference in Supplementary Fig. 9).

Comparison of the drivers between GCMs shows relatively high agreement in the Americas and Africa (Fig. 4e). In Europe and Asia, the inter-model variability is higher (no or only two GCMs agree), potentially due to differences in climate inputs and the subsequent impact on river discharge and water availability.

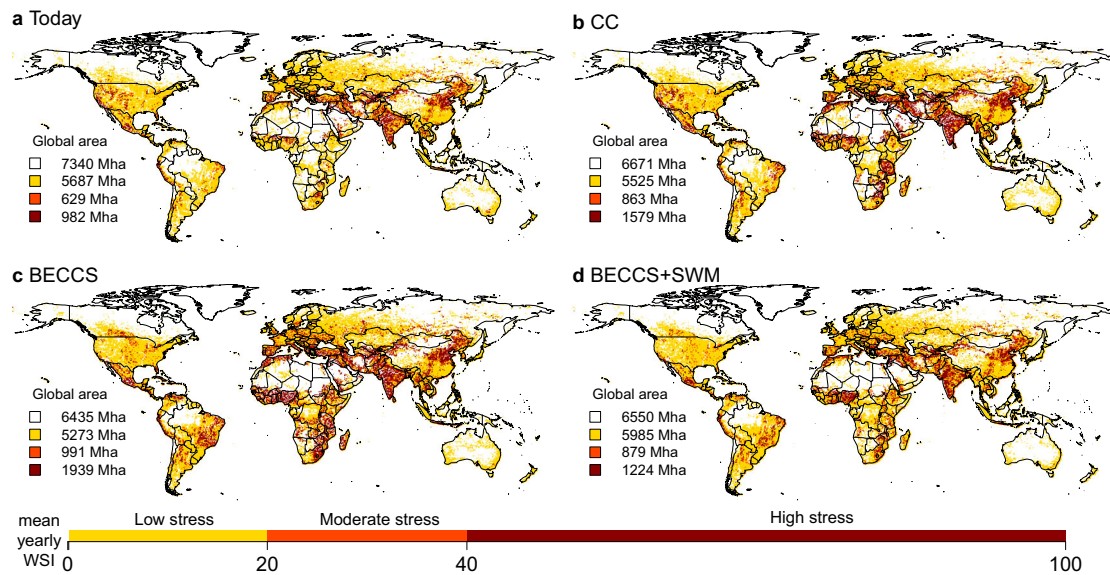

**Fig. 2 WSI simulated under HadGEM2-ES climate forcing for *Today* (2006–2015), and for future scenarios (2090–2099) *BECCS, CC, BECCS+SWM*.** The global numbers refer to the total area exposed to the different degrees of water stress: 0–0.1% (*no stress*), >0.1–20% (*low stress*), >20–40% (*moderate stress*), >40–100% (*high stress*).

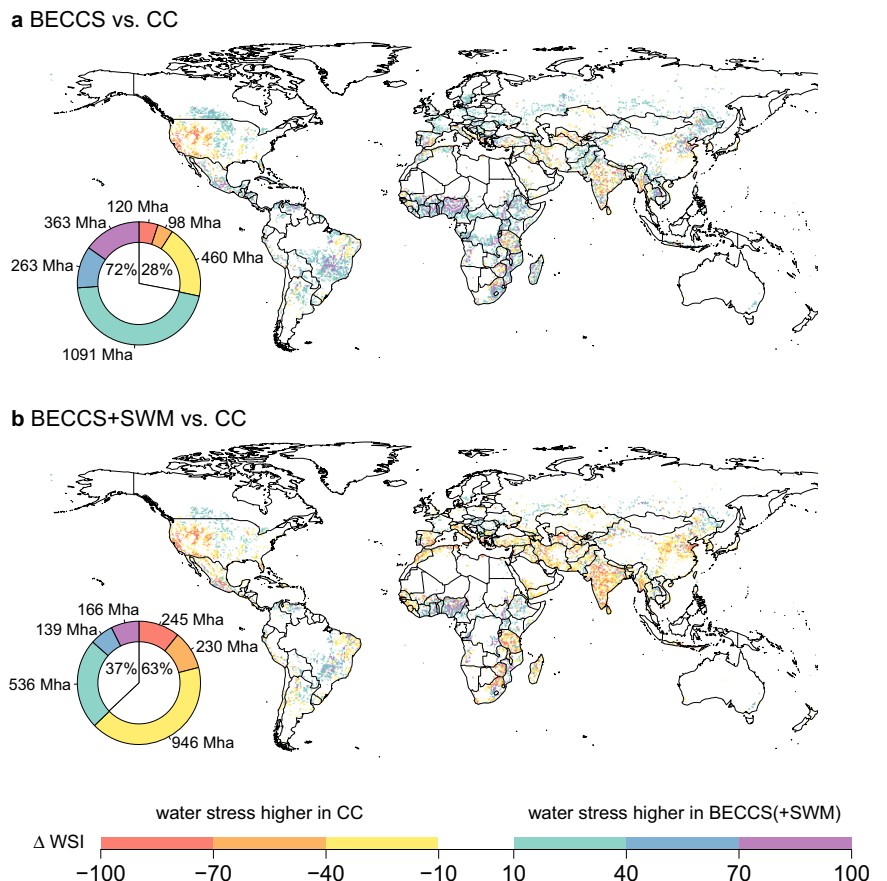

**Fig. 3 Differences in water stress between scenarios BECCS(+SWM) and CC.** Shown are differences in mean yearly WSI values (percentage points) among the different scenarios (here, under HadGEM2 climate forcing, 2090-2099 average). Pie diagrams show the total global area showing a certain (respectively colored) difference.

**Potentials of sustainable water management**. While these results suggest that irrigation for BECCS will lead to stronger increases in water stress than climate change, both globally and regionally, efforts of EFR protection and advanced on-field water management could potentially moderate the effect of irrigated biomass plantations. The respective simulations (scenario *BECCS+SWM*) indicate a strong reduction of the global area under high water stress to 1224 (1167–1327) Mha. The global population under high water stress is

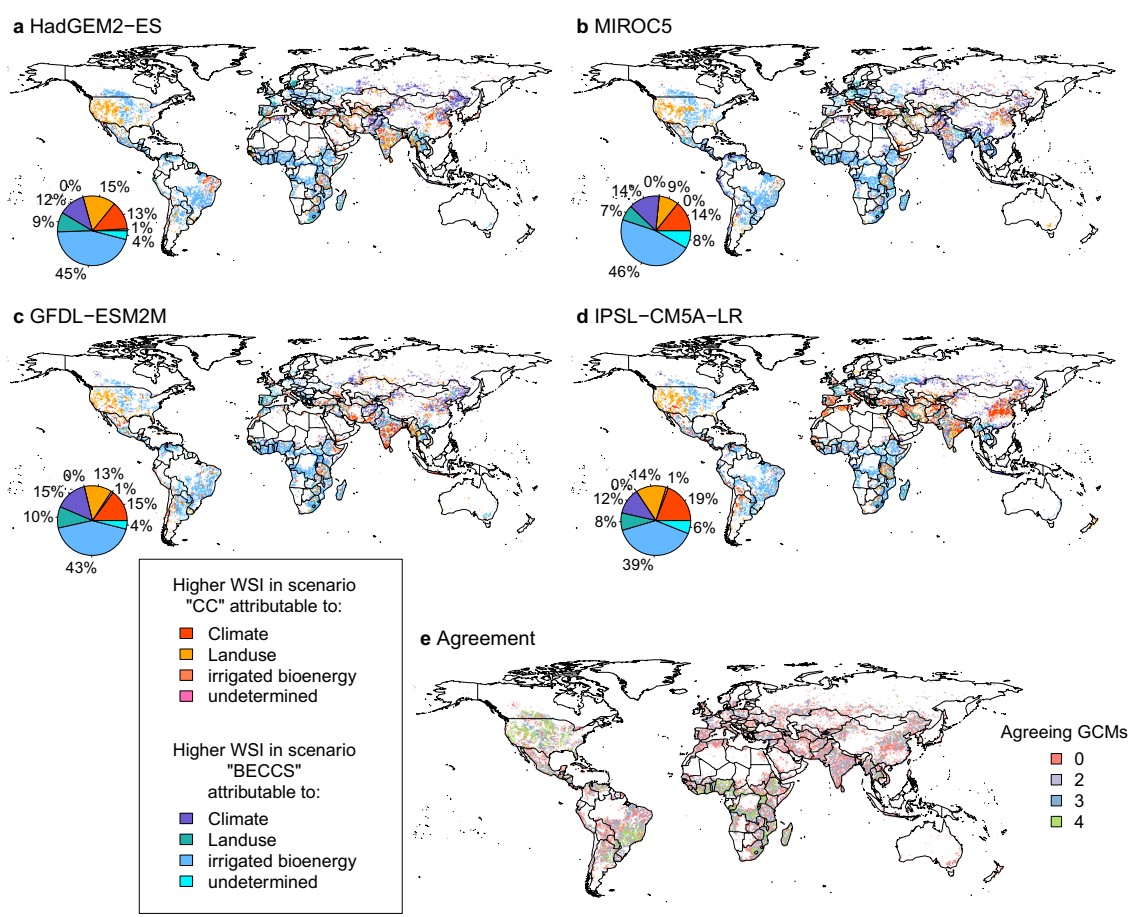

**Fig. 4 Attribution of main driver explaining differences in water stress between the scenarios *BECCS* and *CC*. a–d** Higher water stress in *BECCS* is indicated by blueish colors, the opposite in reddish colors. Drivers are attributed by factorial simulation experiments keeping either land use, climate, or irrigation on biomass plantations constant (see Methods—attribution of drivers for water stress differences). The global area shares of each category are displayed to the bottom-left of each map. **e** Number of GCMs that agree on the attributed driver in a grid cell.

limited to 3.66 (3.47–3.85) billion people. Both area and population in *BECCS+SWM* are reduced to below the values derived for the *CC* scenario (Fig. 1). Also the globally aggregated area under increased water stress would be lower (reduction from 72% to 37%; Fig. 3b), indicating that this scenario globally leads to lesser water stress compared to a scenario with stronger climate change and no bioenergy (maps for all GCMs: Supplementary Fig. 2). This demonstrates that irrigation for BECCS, accompanied by policies directed toward more sustainable and effective freshwater use (here, protection of EFRs and improvements in on-farm water use efficiency including on food-producing cropland), could help avoid the aggravation of water stress. However, significant challenges including investment potential and water resource management practices may hamper the implementation of these policies globally. Moreover, there are regions where even these optimal conditions cannot consistently improve water stress conditions (across GCMs) beyond those of *CC* (Eastern USA, parts of South America, parts of Central and Southern Africa, and parts of Central Europa) (Supplementary Fig. 2). Supplementary Fig. 13 illustrates that, despite SWM, irrigation for BECCS is still the main driver, suggesting that water availability does not allow significant human water withdrawals in these regions.

## Discussion

We conclude that climate mitigation via irrigated BECCS (in an integrated scenario based on RCP2.6), assessed at the global level, will exert similar, or even higher water stress than the mitigated

climate change would (in a scenario based on RCP6.0). This confirms (with the exception of the Western United States) results from a previous study for the United States, where irrigated bioenergy plantations were suggested to increase the annual water deficit in comparison to a climate mitigation scenario[28], albeit the study has different assumptions on land use and climate trajectories and uses a very different model. Potential hotspots for future water scarcity due to irrigated bioenergy as previously highlighted by the same authors[29], do not resemble the patterns which we find, suggesting the need for a larger model intercomparison.

Our results also show that globally, the number of people exposed to severe water stress will generally increase due to climate change and expected population growth[3,41]. It is thus imperative to minimize additional water demand in an already highly water-stressed world, considering also the strong regional differences highlighted in this study.

We thus explicate the dilemma that on top of technological as well as socio-economic barriers to large-scale BECCS deployment[19,42,43], the production of required amounts of biomass (and thus NEs) is further challenged due to freshwater limitations (imposing higher water stress). The reduction of biomass productivity through only cultivating rainfed biomass plantations and discouraging irrigation (50 GtC over the century—Fig. 6); however, might make the difference between 1.5 °C and (likely) 2.0 °C scenarios (87 GtC)[44]. This highlights the need to include water availability limitation in integrated assessment scenarios that look at stringent mitigation futures such as a 1.5 °C world, because

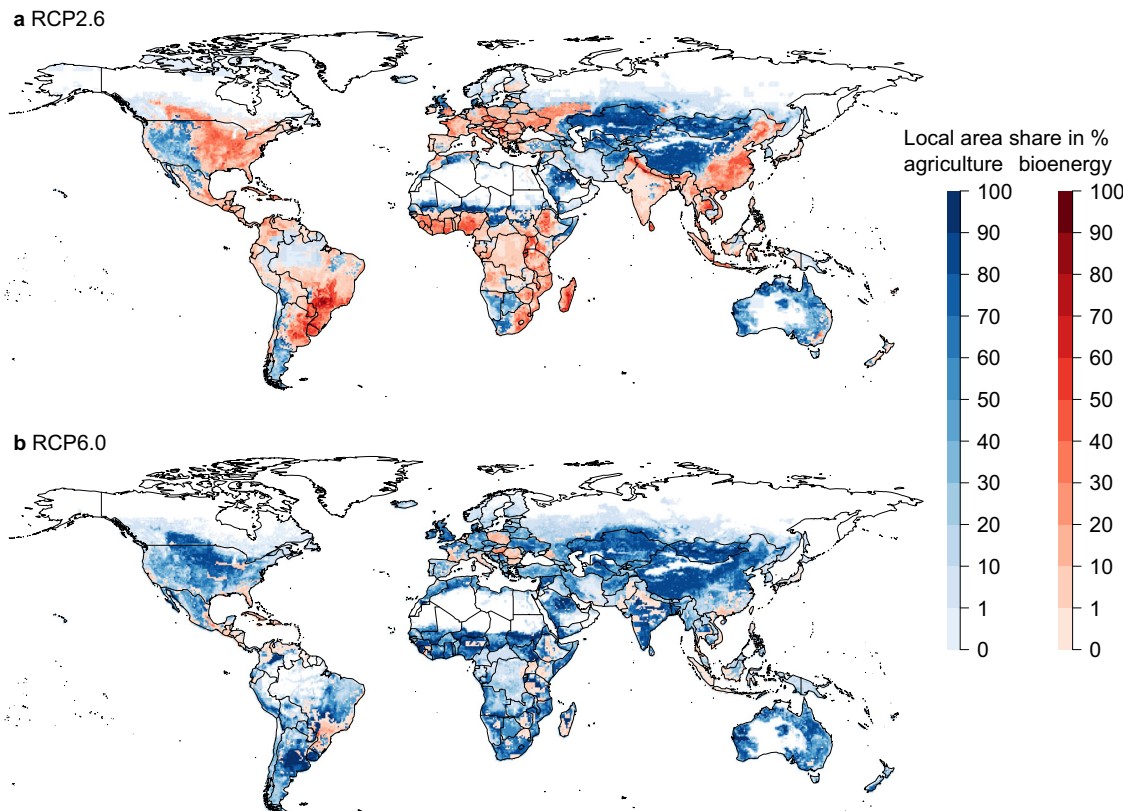

**Fig. 5 Grid-cell area shares of food crops and pastures (blue) overlain with those of bioenergy (red) for 2090–2099 in the associated land use scenarios for RCP2.6 and RCP6.0 (616/29 Mha) in ISIMIP2b for the GCM HadGEM2-ES.** Maps are similar for IPSL-CM5A-LR (623/32 Mha), MIROC5 (592/32 Mha), and GFDL-ESM2M (596/28 Mha) (see Supplementary Figs. 14, 15 and 16).

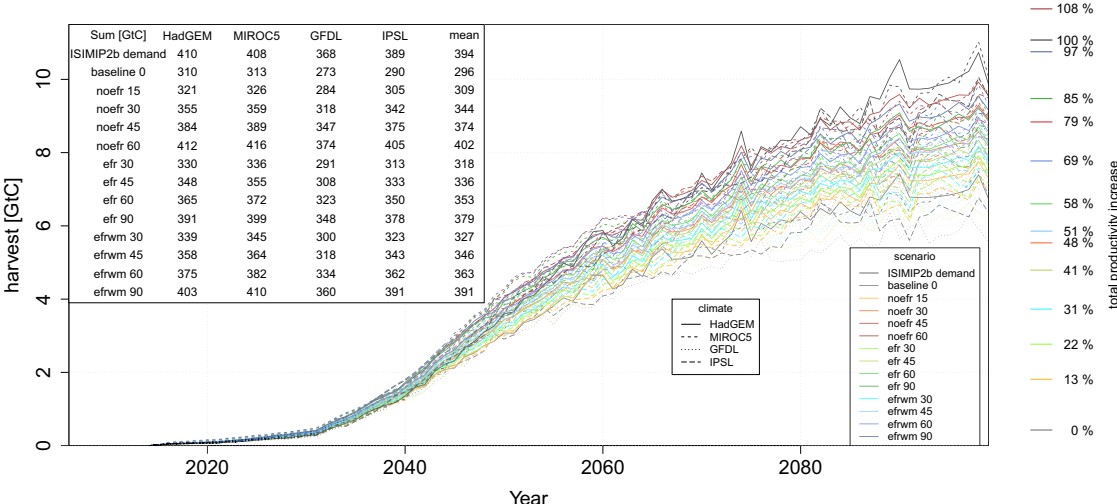

**Fig. 6 Global bioenergy harvest per year for scenarios with 0, 15, 30, 45, 60 or 90% irrigation and no EFR protection (noefr), as well as EFR protection (efr), and EFR protection plus water management (efrwm) for the GCMs HadGEM2-ES, MIROC5, GFDL-ESM2M and IPSL-CM5A-LR.** The ISIMIP2b biomass demand is calculated from the LPJmL yield of the *Baseline* scenario multiplied with the initially assumed productivity increases from MAgPIE. Additionally displayed is the total bioenergy harvest sum over the 21st century of each scenario together with the inter-model mean, which for each scenario is shown on a scale of total productivity increases by technologic change from 0% (*Baseline*) to 100% (ISIMIP2b demand). For further analysis we select scenarios, which can explain ~50% of the productivity increase by the scenario-specific parameters.

it may change the balance of which NE technologies may appear in those scenarios.

Finally, we show that implementation of more efficient water management (in scenario *BECCS+SWM*) could offer a synergistic way out of the water stress dilemma. Achieving this requires the stringent implementation of such methods worldwide[34,45,46], while the required large economic investments (10–20 billion US$ for Africa alone[47]) would also help achieving several SDGs[48].

Growing evidence suggests that next to the direct influence of irrigation on freshwater availability which we account for here, changes in the evapotranspiration regime due to land use change and especially irrigation may also indirectly influence patterns of local rainfall[49,50], and may have remote effects via atmospheric moisture recycling[51,52] or effects on specific and relative humidity[53]. Taking these effects into account requires either a coupled biosphere-atmosphere model or a complex redistribution along atmospheric moisture tracks[54] for each climate scenario, which open up avenues for potential future research.

Water stress is just one aspect of the wide-range of potential impacts of climate change. Similarly, also every technology designed to avoid climate change will entail (potentially not yet known) side effects, which can even be beneficial in some regions but detrimental elsewhere. In this regard, more holistic analyses of the consequences of mitigation portfolios are required that take into account all dimensions of the complex Earth system.

## Methods

**The dynamic global vegetation model LPJmL.** All simulations are conducted with the process-based Dynamic Global Vegetation Model LPJmL[36,55]. The global land surface is separated into 67420 cells from a $0.5° \times 0.5°$ global grid. Daily terrestrial carbon fluxes for establishment, growth and productivity of natural vegetation and agriculture on managed land[56] are simulated dynamically based on climatic conditions. Hydrological processes consider blue and green water fluxes, connected by a river routing network including dams and reservoirs[57–59]. Sowing dates for 12 crop functional types plus a group of other annual and perennial crops are dynamically calculated[60] and calibrated to match national yield statistics[61].

Additionally, pastures and two groups of second-generation bioenergy crops (woody and herbaceous) are considered. Woody species resemble temperate willows and poplars or tropical *Eucalyptus*, while herbaceous species are parameterized as *Miscanthus* and switchgrass[24,62,63]. Field data were used to evaluate bioenergy yields against[64]. A single water use input for SSP2 (to be used in all our scenarios) is prescribed (ISIMIP2b provided multi-model mean domestic and industrial water withdrawal and consumption generated from the ISIMIP2a varsoc runs of WaterGAP, PCR-GLOBWB, and H08[7]). Agricultural areas can be rainfed or irrigated, based on three irrigation techniques: surface, sprinkler, and drip[65]. To improve water use efficiency, management strategies like mulching, local water storage, and conservation tillage can be applied on a grid-cell level (affecting both cropland and bioenergy plantations)[34].

LPJmL can also restrict water withdrawals for irrigation to sustain Environmental Flow Requirement (EFRs), which are calculated from the mean monthly discharges of the last undisturbed period of 1670–1699, before human land use is introduced. Based on the VMF method[35], 60% [45%, 30%] of the local discharge in low [intermediate, high] flow months are withhold to secure riverine ecosystems. The flow regime of a given month is defined through comparison with the mean annual flow. Intermediate-flow months are defined by a mean monthly flow of >40% and <80% of the mean annual flow, low flow months below, and high-flow months above this range.

Within a grid-cell crops are assumed to compete with bioenergy plantations for irrigation water. So by cultivating irrigated bioenergy in water-scarce regions or by restricting withdrawals based on EFRs, crop yields are reduced. Possible solutions for potential yield losses resulting from these strict sustainability scenarios have been previously discussed[23,66]. In our scenarios with water management, the yield decreases are approximately balanced by more effective water management (see Supplementary Fig. 12), which is also applied to cropland[67]. In our simulations, irrigation water demand, which cannot be met by local surface water availability (or would tap EFRs) can also be fulfilled by available water in neighboring cells. Fossil groundwater resources are not considered, but renewable groundwater resources are included as part of the river discharge (baseflow). Return flows are routed back to the river network.

We acknowledge only using a single simulation model. However, the results reported here are largely controlled by the external climate and land use inputs.

**Climate and land use change scenarios.** For maximum consistency, LPJmL was only forced with input from the ISIMIP2b protocol[33]. This includes daily climate data from four General Circulation Models—GCMs—(HadGEM2-ES, MIROC5, GFDL-ESM2M, IPSL-CM5A-LR), as well as cell-based projections of water use[7] and GCM-specific land use patterns (including both biomass and food crops) based on the land allocation model MAgPIE[68] for RCP2.6 and RCP6.0 based on SSP2[69]. MAgPIE simulations ensure that the food demand required also by the growing population is met, including global trade flows to redistribute products and investment in technological change through which crop productivity can be increased.

LPJmL simulations are performed with an initial spinup of 5000 years of potential natural vegetation (based on preindustrial control-climate) to bring global carbon pools to an equilibrium, followed by 307 years of transient spinup using

ISIMIP2b land use patterns from 1700 to 2006. From 2007 to 2100, land use calculated by MAgPIE follows projections aiming at "fulfillment of food, feed and material demand at minimum costs under socio-economic and biophysical constraints"[70] (see Supplementary Fig. 12 for the development of total crop harvests). It needs to be noted, that downscaling of land use patterns from MAgPIE regions to the ISIMIP2b grid has not been based on local water availability. For stringent mitigation scenarios, taking into account this potential yield decreasing effect of water limitations on irrigated crop locations might however be desirable.

The climate and land use trajectories used in this study serve as internally consistent scenarios representing a world with limited climate change through large-scale BECCS, which is compared with a strong climate change world. The ISIMIP2b framework is unique for preparing these internally consistent scenarios. Future versions might include more detailed and higher resolution atmospheric as well as land-system processes (e.g. effects of moisture recycling or sea level rise), and potentially even include complex process couplings which could explain global tipping points[71]. Resilience to complete deforestation, for example, depends on the region and also on the timing of the forest loss[72]. Including such processes would eventually also allow for even more detailed water stress analyses.

Our results are valid for comparing water stress in the two given climate scenarios with approximate GMT increases (inter-model mean rounded to the closest half-degree) 1.5 °C and 3 °C in 2100[33] (RCP2.6—1.68 °C and RCP6.0—3.15 °C) and should be understood as such. Any deviation from the given trajectories (faster/slower emission reductions than envisioned, the crossing of tipping points, or newly discovered Earth system behavior) would require the analysis of the corresponding data.

**Determining the bioenergy irrigation amount.** The ISIMIP2b protocol considers bioenergy plantations for means of energy generation and to realize NEs by BECCS (Fig. 5). Due to land scarcity and potential productivity increases through irrigation, BECCS is likely going to be irrigated to a substantial degree[24,32], however in ISIMIP2b irrigation for bioenergy plantations in the land use scenarios is not included. Instead, the land use projections are based on increasing productivity on cropland due to technological change[73], which can be invested in, but does not have any effect on the plant physiology (e.g. higher water demand through development of genetically modified cultivars with higher leaf area index). LPJmL does not include such technological change and yield increases require additional irrigation or water management efficiency improvements. Therefore the simulated yields on the given land use patterns are lower than what was initially assumed for ISIMIP2b. We call this scenario *Baseline* (no irrigation of bioenergy plantations).

Since we focus on quantitative effects of irrigated bioenergy plantations as a productivity increasing management option, we estimated the amount of irrigation, which could reproduce the initially assumed bioenergy harvests to stay consistent with SSP2 and RCP2.6, (under the given water policy and management conditions). We thus performed a sensitivity analysis by equipping a fraction of the bioenergy plantation area share per grid cell from 0% to 60% in 15% steps (irrigation level) and then focused on those scenarios in our analysis, which could explain ~50% of the additional bioenergy productivity increases over the 21st century in the ISIMIP2b demand compared to our baseline scenario with only rainfed bioenergy plants (Fig. 6). The remaining 50% were assumed to be met by technology improvements, which do not have a direct effect on the water cycle (e.g. more efficient usage in labour or capital). The irrigation level that matched this criterion best, is 30% (*BECCS*). For scenario *BECCS+SWM*, the irrigation level had to be increased to 45%, due to the withdrawal restrictions for environmental flow protection (for scenario overview see Table S1).

Since the irrigation level was applied globally to all bioenergy grid cells, it also introduced irrigation to cells with low local water availability. Withdrawal restrictions in the *BECCS+SWM* scenario then effectively turn the cells bioenergy plantations to pure rainfed again.

The employed land use patterns for agriculture and bioenergy as a result of a global optimization would be different if irrigated bioenergy plantations had not been excluded in the ISIMIP2b protocol[70]. Reaching the same biomass harvest with irrigation potentially requires less plantation area, which could be substituted with other crops. This motivates continued research in defining sustainable regional specific irrigation thresholds and locations based on our water stress maps and a full integration of EFRs and irrigation related parameters into current integrated assessment models.

The ISIMIP2b protocol already includes agricultural residues as additional biomass source for BECCS or bio-fuel production[14]. Recent studies suggest that there might be an additional potential for utilization of organic wastes, which could reduce the raw biomass demand, and thus reduce land or water requirements[74].

**Water stress index WSI.** The water stress index (WSI) is computed individually for each grid cell on a monthly basis as a 10 year average percentage of human water use (withdrawals) compared to total river discharge (which includes renewable groundwater)[26].

$$\text{water stress index} = \frac{\text{domestic} + \text{industrial} + \text{irrigation water use}}{\text{total discharge}} [\%] \quad (1)$$

From the monthly values, we calculate a mean yearly water stress as the main WSI indicator for this study. In the supplementary information, we perform the same

analysis also with cell-based maximum water stress (Supplementary Figs. 3–5, 17), as the water stress of the mostly stressed month (see Supplementary Fig. 18 for a map of these months in scenario BECCS).

**Attribution of drivers for water stress differences**. The differential water stress maps (Fig. 3) show in which of the two compared scenarios (BECCS or CC) the stress is higher, but they do not explain what the driver for this is. Generally, it could be due to the differences in climate input, land use patterns, or the amount of bioenergy irrigation. To perform the attribution, we analyze six scenarios, where pairs of them only differ in one regard (climate input, land use patterns, or irrigated bioenergy extent). We compare the WSI in these three pairs (CCdiff, LUdiff, and IBdiff). CCdiff is composed of two simulations with climate from RCP6.0 and RCP2.6, but the same RCP2.6 land use without irrigated bioenergy to analyze the climate change contribution. LUdiff is based on two simulations with land use from RCP6.0 and RCP2.6 without irrigated bioenergy, but the same climate from RCP2.6 for the land use contribution. IBdiff is calculated from two simulations with irrigated and non-irrigated bioenergy land use from RCP2.6 and the same RCP2.6 climate from RCP2.6 to quantify the irrigated bioenergy component. If a grid cell shows higher water stress in the BECCS scenario, and the absolute of IBdiff is more than 20% higher than that of CCdiff and LUdiff, we mark the cell as Higher WS in scenario BECCS attributable to: irrigated bioenergy (and similar for the other 2 cases). Should 2 or 3 drivers apply at the same time (the differential stresses CCdiff, LUdiff and IBdiff are similar), we mark the cell as undetermined.

## Data availability

Data supporting the main findings of this study are available via https://doi.org/10.5281/zenodo.4297953. Further supplementary data are available from the corresponding author on request.

## Code availability

Model code and analysis scripts are available via https://doi.org/10.5281/zenodo.4297953. Further data are available from the corresponding author on request.

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

## Acknowledgements

This study was funded by the CE-Land+ project of the German Research Foundation's priority program SPP 1689 "Climate Engineering – Risks, Challenges and Opportunities?". Part of the research was developed in the Young Scientists Summer Program at the International Institute for Applied Systems Analysis, Laxenburg (Austria) with financial support from the German National Member Organization (Association for the Advancement of IIASA). PG and YW are financially supported by EUCP (European Climate Prediction System) project funded by the European Union under Horizon 2020 (Grant Agreement: 776613). We thank Miodrag Stevanović for providing the technological change factors underlying the land use scenarios applied. All figures were created using R[75], maps are based on the R package "maps"[76].

## Author contributions

F.S. designed the study with input from S.T., P.G., and D.G. F.S. performed all simulations, analyzed the results, created the figures and prepared the manuscript. W.L., Y.W., D.G., S.T., and P.G. contributed to the interpretation of the results. All authors provided critical feedback and helped shape the research, analysis, and manuscript.

## Funding

## Competing interests

The authors declare that they have no conflict of interest.
