## [Peer Review File · Nature Communications]

REVIEWER COMMENTS

Reviewer #1 (Remarks to the Author):

The paper quantitatively assesses the possible side effects of BECCS on water stress by decoupling related drivers (irrigated bioenergy, climate, land use patterns) using global model simulations. This is a well written and important study providing valuable information on trade-offs between reducing carbon emissions and impacts on water. However, I have few concerns that need to be addressed before the paper is accepted.

#1. The current work is based on earlier work by the same authors (Stenzel et al. 2019), where they have already assessed the availability of freshwater, estimated BECCS water demand for various scenarios, and assessed the degree to which different water management will help to minimize water withdrawal. Thus, the bulk of the result is based on what has been achieved in this earlier work. In the current work, what is being done is to divide the water withdrawal by the water availability to get the water scarcity. I wonder if this little additional work makes it worthwhile to be published in a journal with such high impact as *Nature Communication*. Please state explicitly at the beginning what the added value of the new work is to what you already have done earlier.

#2. The study ignored the significant feedback mechanism between an increased irrigation area and local rainfall reduction as observed in Nebraska (Szilagyi 2018, Szilagyi and Franz 2020), East Africa (Alter et al. 2015), and India (Zeng et al. 2017). In addition, possible rainfall increase in downwind areas from irrigated areas may be high as a result of moisture recycling (Harding and Snyder 2012, Pei et al. 2016, Van der Ent et al. 2010). Without taking into account the feedback mechanism and the recycling of moisture, the study theoretically overestimates the level of water scarcity in certain places and underestimates in others. This would impact the total number of people exposed to water scarcity under various scenarios. I propose that the authors should account for these feedbacks and for the recycling of moisture.

#3. It is not clear from the document to what extent food demand has been met under various scenarios. From Figure S13-S15, for scenario RCP 2.6, large areas of the global land appear to be allocated to bioenergy production. It makes me wonder if the remaining cropland will be enough to grow the required food. Please show the food production (can be in calorie terms) under various scenarios. In addition to the physical scarcity of water that would have a direct effect on food security, competition from bioenergy for the limited water and land would probably increase the food shortages, especially in low-income countries in the south. I encourage the authors to add this topic to the discussion.

References

- Stenzel, F., Gerten, D., Werner, C. and Jägermeyr, J. (2019) Freshwater requirements of large-scale bioenergy plantations for limiting global warming to 1.5 °C. *Environmental Research Letters* 14(8), 084001.
- Szilagyi, J. (2018) Anthropogenic hydrological cycle disturbance at a regional scale: State-wide evapotranspiration trends (1979–2015) across Nebraska, USA. *Journal of Hydrology* 557, 600-612.
- Szilagyi, J. and Franz, T.E. (2020) Anthropogenic hydrometeorological changes at a regional scale: observed irrigation–precipitation feedback (1979–2015) in Nebraska, USA. *Sustainable Water Resources Management* 6(1), 1.
- Alter, R.E., Im, E.-S. and Eltahir, E.A.B. (2015) Rainfall consistently enhanced around the Gezira Scheme in East Africa due to irrigation. *Nature Geoscience* 8(10), 763-767.
- Zeng, Y., Xie, Z. and Zou, J. (2017) Hydrologic and Climatic Responses to Global Anthropogenic Groundwater Extraction. *Journal of Climate* 30(1), 71-90.

Harding, K.J. and Snyder, P.K. (2012) Modeling the Atmospheric Response to Irrigation in the Great Plains. Part II: The Precipitation of Irrigated Water and Changes in Precipitation Recycling. *Journal of Hydrometeorology* 13(6), 1687-1703.

Pei, L., Moore, N., Zhong, S., Kendall, A.D., Gao, Z. and Hyndman, D.W. (2016) Effects of Irrigation on Summer Precipitation over the United States. *Journal of Climate* 29(10), 3541-3558.

Van der Ent, R.J., Savenije, H.H.G., Schaeffli, B. and Steele-Dunne, S.C. (2010) Origin and fate of atmospheric moisture over continents. *Water Resources Research* 46(9), W09525.

Mesfin Mekonnen

Reviewer #2 (Remarks to the Author):

The paper discusses the tradeoff between an important negative emission technology (bioenergy with carbon capture and storage) and water scarcity in a global context. The paper is well written, and touches on a very important topic given the growing interest in negative emission technologies and the growing desire to achieve 1.5 C target without jeopardizing SDG goals such as achieving water security.

The main concern is that the argument that climate change mitigation policies could lead to massive water scarcity issues was documented in Hejazi et al 2014 [Hejazi, M. I., Edmonds, J., Clarke, L., Kyle, P., Davies, E., Chaturvedi, V., Wise, M., Patel, P., Eom, J., and Calvin, K.: Integrated assessment of global water scarcity over the 21st century under multiple climate change mitigation policies, *Hydrol. Earth Syst. Sci.*, 18, 2859–2883, <https://doi.org/10.5194/hess-18-2859-2014>, 2014.]. The authors state: "To our best knowledge, this is the first global study contrasting water stress in a strong climate change scenario with limited BECCS (leading to a rise of global mean temperatures of 3 C) and a lower-warming scenario with significant BECCS contribution based on a bioenergy demand trajectory designed to limit global warming to well below 2 C." Obviously, this work is unique in many ways and differ in its approach from the work of Hejazi et al., however the finding that climate change mitigation could increase scarcity was documented previously.

On the experimental design, the authors have done a great job explaining their scenarios and doing several sensitivity analyses to shed light on how their results might change under different assumptions, which is very valuable. On the choice of the default scenarios, I find the argument to assume 0% irrigation under the CC scenario to be an extreme assumption. To allow for a fair comparison between the CC and BECCS scenarios and the effect of expanding biomass area from 30 Mha to 600Mha, I would suggest that irrigation assumption should remain the same across both scenarios (30%).

On a similar topic, I appreciate the effort to include the SWM scenario as part of the analysis. I found the assumption that EFR to be only enforced under that SWM scenario to be somewhat unusual. I would argue that EFC should be assumed under all scenarios and should be included as one of the terms on the numerator of equation 1 (lines 441-442).

[Lines 99-105] need to explain what 's meant by the 'yearly mean' and ' in at least one month'. Also, it is not clear what the two numbers (20/88 and 60/101) refer to? I would suggest caveating the monthly analysis or simply to focus on the annual analysis. The monthly estimates depend heavily on water management assumptions and how reservoir operations are simulated now and into the future. I am assuming those were not considered since they were not mentioned at all.

[lines 106-116] An important sensitivity that the authors don't account for is how the future land use

scenarios are downscaled to the grid resolution. This could have implication based on assumptions about extensification vs intensification, and if there were some water availability (soil moisture) suitability measures taken into consideration. The answer might be that such alternative realizations of such scenarios don't exist. If that's case, then this should be at least acknowledged.

One recommendation that could be emphasized in the paper is to make the case for the need to include water availability limitation in IA scenarios that look at stringent mitigation futures such as a 1.5 C world. This may change the balance of which NE technologies may appear in those scenarios.

[lines 370] 'water use for HIL' what is meant by HIL? I don't think it was ever defined in the paper. Also, it would be important to clarify if the water use for other sectors was the same for all three scenarios in the paper, or if they were consistent with evolution of the energy/electricity sector for example under an RCP6 vs RCP2.6 scenarios. I am not asking for the authors to repeat the work with different water use scenarios for the non-ag sectors, but to at least acknowledge that this is not accounted for here.

How does the model handle unmet flows, return flows, and groundwater pumping? It would good to explain these briefly in the Supp methods section.

Reviewer #3 (Remarks to the Author):

The paper uses the vegetation and water balance model LPJmL to assess the level of water stress in three scenarios: 1) Limited use of bioenergy with carbon capture and storage (BECCS) which is projected to result in a +3 °C warming by 2100 (a radiative forcing 6.0 w/m², RCP6.0); 2) Significant use of BECCS projected to result in limiting warming to 1.5 °C by 2100 (RCP2.6), and; 3) Significant use of BECCS combined with a higher degree of sustainable water management. The LPJmL model is forced by input data from four general circulation models' SSP2 (middle-of-the-road shared socioeconomic pathways) scenarios.

While the paper finds that water stress increases in all scenarios, the more interesting finding is that the 1.5°C significant BECCS scenario results in water stress at similar levels, even possibly slightly higher, as the 3.0°C limited BECCS scenario. The paper also finds that the significant BECCS with increased levels of sustainable water management scenario would (perhaps unsurprisingly) lessen water stress compared to the significant BECCS without strong sustainable water management scenario.

In short, it is my opinion that the paper is novel, justifies its conclusions (yet require more context for how to understand the findings through clarifying model limitations), addresses an important topic, and will most likely attract a large readership.

In terms of novelty, the paper is promising. Not much has been done in this field. I would ask the authors to possibly relate to recent finding by Bin Hu et al. (2020), if relevant. Please consult "Can bioenergy carbon capture and storage aggravate global water crisis?" in Science of The Total Environment, 714, 136856. <https://doi.org/10.1016/j.scitotenv.2020.136856>.

I would also have appreciated a more exhaustive explanation of the different approaches in the paper under review and in the article by Yamagata et al. "Estimating water–food–ecosystem trade-offs for the global negative emission scenario (IPCC-RCP2.6)" from 2018, which is cited in the paper (<https://doi.org/10.1007/s11625-017-0522-5>).

Both papers arrive at similar conclusions to those suggested by the authors, i.e. that BECCS may elevate water stress. It would be nice if the authors could make explicit why the approach chosen in the reviewed nature-paper is novel and complementary to, or even preferred over, the approaches of Hu et al. (2020) and Yamagata et al. (2018).

The paper has potential to influence thinking in the field, especially if the claims are made more convincing through improved clarity on a number of key model assumptions/limitations. In my view, the paper would be further strengthened if it also included a +1.5 °C (RCP2.6, SSP2) Limited BECCS scenario, to assess water stress levels when in a scenario compatible with limiting global warming to +1.5°C with only limited BECCS (such as in the IPCC SR1.5 P1 scenario). Let me expand on these proposals, i.e. to increase transparency/clarity and the rationale for a +1.5 °C Limited BECCS scenario.

On clarity, I would first like to state that the paper, in terms of language, is well written and understandable. Substantial language edits are unnecessary. When I seek more clarity, it is rather due to the complexity of the models involved. I seek further guidance on a number of issues, starting with feedbacks and uncertainty:

1. How do you deal with the fact that cloud formation and rainfall are two aspects that are highly uncertain in a changing climate? Maybe you've addressed this already in the text (and I've failed to understand it). If you have, could you please make it more explicit and understandable to a general audience? If you haven't addressed it, I think it deserves attention. To what extent does it affect how to understand your results? And is there, for example, a difference in certainty of cloud formation and rainfall patterns in a climate with +1.5 °C average surface warming (much closer to today's approx. +1.0 °C warming) compared to +3.0 °C?
2. Does your model explicitly account for tipping points? Or is this perhaps implicitly accounted for in the forcing-data for your model?
 - a. If yes, at what temperature levels are crucial tipping points expected to be triggered? What kinds of bifurcated hysteresis are represented in your scenarios? Does the irreversibility of perturbed new climates, after tipping points, affect your conclusions about water stress in the Limited BECCS +3.0 °C scenario? As a consequence of irreversible climate change, would water stress (understood partially as an impact of climate change) also be irreversible? And could water stress in the Significant BECCS and Significant BECCS+SWM scenarios perhaps be reversible, if the climate forcing is reduced beyond +1.5 °C? Is this important to note?
 - b. If no, what does this mean for how to understand your results? What does the literature on tipping points have to say about triggering such in a 3.0 °C world? Would +3.0 °C lead to runaway climate change or irreversible system change that makes water stress even worse than indicated by your findings? Would this affect your conclusions about the Significant BECCS with or without SWM scenarios?
3. Does your model/do the model forcing include/represent reversible feedbacks, i.e. other than cloud formation and irreversible feedbacks triggered through crossing tipping points, that would severely affect hydrology?
4. A specific form of impact of climate change that I would guess affects water stress quite substantially is sea level rise. Do you account for this? What is the difference of sea level rise in a

+1.5 °C world compared to a +3.0 °C world? Loss of land and salt water penetration? While sea level rise is linked to both point 2 and 3 above, I thought I'd mention it separately. I've seen some quite remarkable estimates of loss of land and forced displacement in the wake sea level rise induced by +3 °C warming, but I'm not qualified to assess their validity.

5. Are all three scenarios that you explore attached with the same level of likelihood of achieving 1.5 °C and 3.0 °C?

6. I assume that the Significant BECCS and Significant BECCS+SWM scenarios both are assumed to lead to 1.5 °C (with the same level of likelihood), i.e. that they result in roughly the same levels of concentration of greenhouse gases in the atmosphere? This is unclear, i.e. particularly the extent to which the Significant BECCS+SWM scenario is expected to lead to 1.5 °C. It is clear that both lead to a radiative forcing of 2.6 w/m², but it remains unclear to me (as a reader with limited understanding of climate modelling) if this automatically means that they are expected to lead to 1.5 °C or if the forcing of the LPJmL model with input from four global circulation models means that you import different assumptions about climate sensitivity in the different scenarios or somehow use different likelihoods.

Please review my comments above in context of my limited understanding of climate modeling, whether using GCMs or IAMs. I also note that you, on lines 209-210, introduces a form of disclaimer that the Earth system is much more complex than your model. That your model is not as complex as the Earth system is quite obvious, but I do think that you may want to explicitly address some of the consequences of (the completely necessary) failure to capture all of the Earth system's complexity for how to understand your findings. I don't ask of you to quantify responses to all of the issues I raise above, as this would probably not be doable within your model framework. But I do ask you to reflect, in qualitative terms, on their potential significance for how to interpret your results, to the extent that you find the comments relevant.

On clarity, again, I also seek further guidance on issues relating to assumptions made regards BECCS:

7. Not all biomass-energy is coupled with CCS in IAM scenarios. Are you actually investigating scenarios of irrigated biomass plantations (regardless if the biomass is used in operations with or without CCS), using irrigated biomass as a proxy for BECCS? The article is ambiguous on this point, but I sometimes feel like biomass is made equal to BECCS. I'm sure you don't, but it deserves further clarification. This connects to your proposed title for the paper. The title should probably signal the importance of BECCS that you stress throughout the main body text. In its present form, the title seems slightly off target, unless, of course, what you actually do is to investigate biomass—water stress linkages rather than biomass-limited-to-BECCS—water stress linkages. If it is actually the former, you should make this much more explicit in the paper.

8. You discuss the application of BECCS in production of electricity and liquid biofuels. These are promising activities that already emits high levels of biogenic CO₂. But is waste to energy included represented by your model/forcing? Both electricity (from incineration of waste) and liquid biofuels (from agricultural residues) are promising candidates for BECCS and would not, I guess, increase demand for bioenergy (other than, perhaps, to fuel the energy penalty for carbon capture and transport from waste incineration and for transport in bioethanol production). It could complement dedicated biomass boilers and cofired biomass-fossil plants with CCS.

9. Similarly, does your approach account for the BECCS potential in production of gaseous biofuels and in pulp and paper? These activities have a high potential for BECCS, again without BECCS being the main driver behind biomass demand (the demand is for fuel and paper, with CCS being a potential

technology to reduce emissions linked to these activities to the extent that they can become negative; the world population, however, would likely not by more paper just to capture and store biogenic CO₂ in order to resolve climate change).

On the importance of a 1.5 °C with limited BECCS scenario:

10. The above leads me to a second line of thought, which if you agree to it, would require more revisions. In my view, it would, however, also substantially improve your paper.

On line 50-51, you seem to assume that BECCS drive demand for biomass. But, to my knowledge, what happens if you exclude BECCS from models' technology portfolios is that biomass demand increases. This is an effect of a need to phase out fossil fuels earlier than would otherwise be necessary, leading to an increased demand for liquid biofuels to decarbonize the transport sector. For the same temperature goal, say 1.5 °C, BECCS may thus serve to lessen rather than elevate the demand for biomass in IAM scenarios.

Would it not, therefore, be suitable to compare a +1.5 °C Significant BECCS scenario not only to a +3.0 °C Limited BECCS scenario but also to a +1.5 °C Limited BECCS scenario?

If explored in detail, a quite acceptable compromise between fully taking this comment into account and produce a completely new scenario, and to completely neglect this comment, would be to use previous literature on this topic reflect upon how to interpret your findings in view of the likely result of producing a +1.5 °C Limited BECCS scenario (without actually doing so).

Smaller revision points:

Line 25: Consider clarifying that "will" is conditional to the model design. "Will" seems to be a strong word in context of scenarios, even if preceded by the qualifier "suggest".

Line 26: Consider rephrasing to improve the clarity of the sentence. You could express this as something along the lines of: "The increased water stress induced by irrigated BECCS may even exceed avoided water stress achieved by using BECCS to limit global warming to 1.5 °C instead of 3.0 °C".

Line 23-26: In making revisions on line 25 and 26, consider breaking the long sentence on line 23-26 into two sentences to facilitate the reading.

Table 1, second row: The asterisk is unexplained, lacks a "sibling" that explains what is meant by it.

Line 81: Clarify if you study scenarios or a scenario.

Line 101-105: Unclear. I think that it would be better to express the relative (percentage) increase reported on line 103 in absolute terms. As you do on line 117, for example. You could then also express this as the share of global population in 2090–99. Or as a percentage increase of the absolute numbers. In the current format, it is hard to understand what you mean when you say that the you've included the effect of population increase in the. It only becomes clear when studying the numbers in Figure 1.

Figure 1: Consider facilitating the reading of figures when viewed in greyscale. This shouldn't be too

hard for Figure 1, at least, even if worse (and therefore should probably be ignored) for the rest of the figures.

Line 160: It is hard to make sense of the statement that in inter-model variability, one CGM is in agreement. Do you intend for this to mean that none of the four GCMs agree in those cases? Or do you actually mean to say "... (two or three GCMs agree) ..." rather than "... (one or two GCMs agree) ..."? In any case, maybe you could make this clearer to the reader.

I really enjoyed reading your paper and hope that some of my comments will be helpful in guiding revisions.

Best, Mathias Fridahl, Linköping University

REVIEWER COMMENTS

Reviewer #1 (Remarks to the Author):

The paper quantitatively assesses the possible side effects of BECCS on water stress by decoupling related drivers (irrigated bioenergy, climate, land use patterns) using global model simulations. This is a well written and important study providing valuable information on trade-offs between reducing carbon emissions and impacts on water. However, I have few concerns that need to be addressed before the paper is accepted.

Answer:

We thank Dr. Mekonnen for his review and his assessment that our study provides a valuable addition to the current literature. Below is a point by point response to the comments:

#1. The current work is based on earlier work by the same authors (Stenzel et al. 2019), where they have already assessed the availability of freshwater, estimated BECCS water demand for various scenarios, and assessed the degree to which different water management will help to minimize water withdrawal. Thus, the bulk of the result is based on what has been achieved in this earlier work. In the current work, what is being done is to divide the water withdrawal by the water availability to get the water scarcity. I wonder if this little additional work makes it worthwhile to be published in a journal with such high impact as Nature Communication. Please state explicitly at the beginning what the added value of the new work is to what you already have done earlier.

Answer 1:

The approach in Stenzel et al., 2019 is quite different, and actually we did neither use the biomass plantation patterns simulated therein, nor adopt its methodology. In that paper, we assumed current cropland to stay constant while bioenergy plantation areas were gradually added up to an extent needed to limit mean global warming to 1.5°C. Also we did not analyze a climate change scenario in those rather stylized scenarios. In contrast, the aim of the present study is to compare two consistent future scenarios, both of which provide transient land use (crops and bioenergy), climate, and water use trajectories for the 21st century (according to the ISIMIP2b protocol). Findings are expected to be more robust because we accounted for consistent scenarios rather than the stylized scenario of the 2019 paper. We made this clearer in the Introduction (lines 73-76).

#2. The study ignored the significant feedback mechanism between an increased irrigation area and local rainfall reduction as observed in Nebraska (Szilagyi 2018, Szilagyi and Franz 2020), East Africa (Alter et al. 2015), and India (Zeng et al. 2017). In addition, possible rainfall increase in downwind areas from irrigated areas may be high as a result of moisture recycling (Harding and Snyder 2012, Pei et al. 2016, Van der Ent et al. 2010). Without taking into account the feedback mechanism and the recycling of moisture, the study theoretically overestimates the level of water scarcity in certain places and underestimates in others. This would impact the total number of people exposed to water scarcity under various scenarios. I propose that the authors should account for these feedbacks and for the recycling of moisture.

Answer 2:

Thank you for pointing to these moisture feedbacks possibly associated with our irrigation simulations, which are indeed not possible to be considered in a stand-alone land surface/biosphere model like LPJmL without direct coupling to the atmosphere. While we agree that such feedbacks can be important regionally, they have not yet been considered in any global hydrological model that provides future projections of irrigation water use, including the state-of-the-art studies conducted within the ISIMIP framework. First steps could be to exchange LPJmL evapotranspiration fluxes with atmospheric moisture transport models, though explicit global projections of moisture tracks have been performed (e.g. by Tuinenburg et al., 2020) only for historical reanalysis climate data (also without considering explicit moisture recycling cascades)

and can thus not directly be incorporated here. Analyzing such fluxes for the future, under conditions of climate change, would require coupled biosphere-atmosphere models as the current moisture transport trajectories (“atmospheric rivers”) may significantly change in the future. That said, we see no feasible option to reproduce the downwind changes in response to irrigation in our current scenario setup, not least because we expect this feedback to also be GCM-specific, which eventually would require to couple LPJmL to each of the GCMs we use climate input from. This certainly requires a larger study on its own.

We therefore decided to expressly acknowledge these limitations and encourage future studies on these effects in the manuscript by adding a respective paragraph to the discussion (lines 228-234).

#3. It is not clear from the document to what extent food demand has been met under various scenarios. From Figure S13-S15, for scenario RCP 2.6, large area of the global land appears to be allocated to bioenergy production. It makes me wonder if the remaining cropland will be enough to grow the required food. Please show the food production (can be in calorie term) under various scenarios. In addition to the physical scarcity of water that would have a direct effect on food security, competition from bioenergy for the limited water and land would probably increase the food shortages, especially in low-income countries in the south. I encourage the authors to add this topic to the discussion.

Answer 3:

Thank you for this important question. We fully agree that future food security is central in debates about the bioenergy-water-food-environment nexus.

The land use input (including areas for biomass production) provided by the ISIMIP2b protocol is based on results from the agro-economic model MAgPIE, which ensures achieving the increasing food demand of a growing population (in our study using trajectories of the SSP2 socio-economic scenario). MAgPIE includes investments in technological change to increase crop productivity and global trade for redistributing agricultural commodities to achieve an optimal (low-cost) production. To clarify this, we extended the description of the land-use input in the “Climate and land use change scenarios” section of the Methods (lines 279-286).

Additionally, we added Figure S12 to the SI, showing that crop harvests grow with increasing population. It also displays that adding irrigation in the BECCS scenario has little effect on harvests compared to the Baseline (the curves are virtually identical). Limiting the irrigation water withdrawals in the BECCS+SWM scenario is approximately balanced by the increased on-field water use efficiency (total crop harvests are between 3% and 5% lower), which is much smaller than the variability induced by the different climate inputs (up to 18%).

Reviewer #2 (Remarks to the Author):

The paper discusses the tradeoff between an important negative emission technology (bioenergy with carbon capture and storage) and water scarcity in a global context. The paper is well written, and touches on a very important topic given the growing interest in negative emission technologies and the growing desire to achieve 1.5 C target without jeopardizing SDG goals such as achieving water security.

Answer:

We thank the reviewer for this generally positive evaluation, and appreciate her/his assessment that we are addressing a very important nexus issue with our research. Below is a point by point response to the comments:

1) The main concern is that the argument that climate change mitigation policies could lead to massive water scarcity issues was documented in Hejazi et al 2014 [Hejazi, M. I., Edmonds, J., Clarke, L., Kyle, P., Davies, E., Chaturvedi, V., Wise, M., Patel, P., Eom, J., and Calvin, K.: Integrated assessment of global water scarcity over the 21st century under multiple climate change mitigation policies, Hydrol. Earth Syst. Sci., 18, 2859–2883, <https://doi.org/10.5194/hess-18-2859-2014>, 2014.]. The authors state: “To our best knowledge, this is the first global study contrasting water stress in a strong climate change scenario with limited BECCS (leading to a rise of global mean temperatures of 3 C) and a lower-warming scenario with significant BECCS contribution based on a bioenergy demand trajectory designed to limit global warming to well below 2 C.” Obviously, this work is unique in many ways and differ in its approach from the work of Hejazi et al., however the finding that climate change mitigation could increase scarcity was documented previously.

Answer 1:

The point that “climate change mitigation could increase water scarcity” globally has been raised before only as part of a modeling study by Hejazi et al. (2014) that shows water scarcity maps for several energy demand scenarios. The novelty of our research is that we directly compare water scarcity between scenarios of strong climate change with those of large-scale BECCS as a supporting mitigation measure to avoid this very climate change. This demonstrates one of the dilemmas of humanity in our efforts to achieve all SDGs: either way water stress is likely to increase, unless sustainable water management is ensured (as a possible way out). We thus perform a multiple-scenario analysis of the implied trade-offs with a process-based biosphere model, which either restricts irrigation water withdrawal to available surface water, constrains it further to meet environmental flow requirements, and/or assumes effective water management. We thus enrich previous findings with a sophisticated analysis using different scenarios that explicitly focus on the various trade-offs, but also point out a possible way forward.

To make this ambition clearer (and also mention the advancements compared to previous studies) we expanded parts of the Introduction (lines 67-73) and the discussion section (lines 207-209).

2) On the experimental design, the authors have done a great job explaining their scenarios and doing several sensitivity analyses to shed light on how their results might change under different assumptions, which is very valuable. On the choice of the default scenarios, I find the argument to assume 0% irrigation under the CC scenario to be an extreme assumption. To allow for a fair comparison between the CC and BECCS scenarios and the effect of expanding biomass area from 30 Mha to 600Mha, I would suggest that irrigation assumption should remain the same across both scenarios (30%).

Answer 2:

We agree and have increased the bioenergy irrigation level also to 30% for the CC scenario, which did not change the overall results substantially.

3) On a similar topic, I appreciate the effort to include the SWM scenario as part of the

analysis. I found the assumption that EFR to be only enforced under that SWM scenario to be somewhat unusual. I would argue that EFC should be assumed under all scenarios and should be included as one of the terms on the numerator of equation 1 (lines 441-442).

Answer 3:

We perceive EFR protection as part of more sustainable water use which is not the case in many regions yet. Only limiting withdrawals for EFR protection without increased on-field water use efficiencies would decrease irrigated food crop harvests and disrupt internal consistency with SSP2. In terms of water stress a “BECCS+EFR“ scenario would be very similar to the “BECCS+SWM” scenario, since both limit absolute withdrawal to the same environmental flow requirements (compare scenarios EFR and WM of Figure 2 in Stenzel et al., 2019).

The scenario BECCS+SWM is thus designed to showcase a rather optimal scenario (in terms of water stress relief). The combined water stress reduction potentials of environmental flow protection and increased on-field water efficiency compared to the pure BECCS scenario are one of our main results.

To make this clear, we added to the motivation for the BECCS+SWM scenario (lines 93-96).

Reference:

Stenzel, F.; Gerten, D.; Werner, C. & Jägermeyr, J.

Freshwater requirements of large-scale bioenergy plantations for limiting global warming to 1.5°C, *Environmental Research Letters*, **2019**, *14*, 084001

4) [Lines 99-105] need to explain what ‘s meant by the ‘yearly mean’ and’ in at least one month’. Also, it is not clear what the two numbers (20/88 and 60/101) refer to? I would suggest caveating the monthly analysis or simply to focus on the annual analysis. The monthly estimates depend heavily on water management assumptions and how reservoir operations are simulated now and into the future. I am assuming those were not considered since they were not mentioned at all.

Answer 4:

Dams and reservoirs were modeled, we now added this to the model description section.

We followed your suggestion to remove the max. month WSI part of the analysis from the main manuscript, which also improves readability of the Results section. Correspondingly, parts of the Introduction and Methods section, as well as Figure 1-4 were redone and previous maps for max. month WSI were moved to the SI.

5) [lines 106-116] An important sensitivity that the authors don’t account for is how the future land use scenarios are downscaled to the grid resolution. This could have implication based on assumptions about extensification vs intensification, and if there were some water availability (soil moisture) suitability measures taken into consideration. The answer might be that such alternative realizations of such scenarios don’t exist. If that’s case, then this should be at least acknowledged.

One recommendation that could be emphasized in the paper is to make the case for the need to include water availability limitation in IA scenarios that look at stringent mitigation futures such as a 1.5 C world. This may change the balance of which NE technologies may appear in those scenarios.

Answer 5:

As the reviewer already anticipated, there is only one available realization of the land use scenarios per GCM. We added a sentence to the land use scenario description in the Methods regarding the limitations of land use downscaling and water availability (lines 292-296).

We also adopted your highly relevant suggestion to highlight the need to look at water availability in IAMs in the discussion (lines 219-222).

6) [lines 370] ‘water use for HIL’ what is meant by HIL? I don’t think it was ever defined in the paper. Also, it would be important to clarify if the water use for other sectors was the same for all three scenarios in the paper, or if they were consistent with evolution of the energy/electricity sector for example under an RCP6 vs RCP2.6 scenarios. I am not asking for the authors to repeat the work with different water use scenarios for the non-ag sectors, but to at least acknowledge that this is not accounted for here.

Answer 6:

We removed the abbreviation HIL (water use for household, industry and livestock). Water use according to the ISIMIP2b protocol, which we follow here, is based on the SSP2 scenario and the same for RCP2.6 and RCP6.0 – we now acknowledge this in the model description (lines 252-255).

7) How does the model handle unmet flows, return flows, and groundwater pumping? It would good to explain these briefly in the Supp methods section.

Answer 7:

We also added this to the model description section (lines 271-275).

Reviewer #3 (Remarks to the Author):

The paper uses the vegetation and water balance model LPJmL to assess the level of water stress in three scenarios: 1) Limited use of bioenergy with carbon capture and storage (BECCS) which is projected to result in a +3 °C warming by 2100 (a radiative forcing 6.0 w/m², RCP6.0); 2) Significant use of BECCS projected to result in limiting warming to 1.5 °C by 2100 (RCP2.6), and; 3) Significant use of BECCS combined with a higher degree of sustainable water management. The LPJmL model is forced by input data from four general circulation models' SSP2 (middle-of-the-road shared socioeconomic pathways) scenarios. While the paper finds that water stress increases in all scenarios, the more interesting finding is that the 1.5°C significant BECCS scenario results in water stress at similar levels, even possibly slightly higher, as the 3.0°C limited BECCS scenario. The paper also finds that the significant BECCS with increased levels of sustainable water management scenario would (perhaps unsurprisingly) lessen water stress compared to the significant BECCS without strong sustainable water management scenario.

In short, it is my opinion that the paper is novel, justifies its conclusions (yet require more context for how to understand the findings through clarifying model limitations), addresses an important topic, and will most likely attract a large readership.

Answer:

We thank Dr. Fridahl for his positive assessment and have addressed his remaining comments in the following point-by-point responses.

In terms of novelty, the paper is promising. Not much has been done in this field. I would ask the authors to possibly relate to recent finding by Bin Hu et al. (2020), if relevant. Please consult “Can bioenergy carbon capture and storage aggravate global water crisis?” in *Science of The Total Environment*, 714, 136856. <https://doi.org/10.1016/j.scitotenv.2020.136856>. I would also have appreciated a more exhaustive explanation of the different approaches in the paper under review and in the article by Yamagata et al. “Estimating water–food–ecosystem trade-offs for the global negative emission scenario (IPCC-RCP2.6)” from 2018, which is cited in the paper (<https://doi.org/10.1007/s11625-017-0522-5>). Both papers arrive at similar conclusions to those suggested by the authors, i.e. that BECCS may elevate water stress. It would be nice if the authors could make explicit why the approach chosen in the reviewed nature-paper is novel and complementary to, or even preferred over, the approaches of Hu et al. (2020) and Yamagata et al. (2018).

Answer:

Hu et al. (2020) derive their results from meta-analyses of existing literature and approximations of global water demands by extrapolating current water use efficiencies (as the mean of country specific literature values but applied globally) for future energy demand scenarios. Their water stress metric is the ratio of total estimated water use for BECCS divided by the globally available water (and they confusingly refer to both 45,500km³ – referencing Oki and Kanae, 2006 – and 63,283 km³/yr – citing Hanasaki et al., 2013a/b).

Our approach is to use a process based model, using site specific parameters to spatially explicitly compare water stress between scenarios. Only as the last step, for Figure 1, we aggregate to global area or population under severe water stress, which should be preferred, since water stress is a local issue and cannot meaningfully be measured just from global values.

Yamagata et al. (2018) calculate the freshwater demand for BECCS as part of a comprehensive modeling study using a combination of the H08 and VISIT models, however they do not analyze water stress as such. They exclusively assume use of non-river water (e.g. groundwater) for bioenergy irrigation. We argue (also in Answer 1 to Reviewer 2) that for a comprehensive comparison of water stress, local water availability should be used as the reference irrigation source. We correspondingly change the respective part of the Introduction (lines 67-70).

The paper has potential to influence thinking in the field, especially if the claims are made more convincing through improved clarity on a number of key model assumptions/limitations. In my view, the paper would be further strengthened if it also included a +1.5 °C (RCP2.6, SSP2) Limited BECCS scenario, to assess water stress levels when in a scenario compatible with limiting global warming to +1.5°C with only limited BECCS (such as in the IPCC SR1.5 P1 scenario). Let me expand on these proposals, i.e. to increase transparency/clarity and the rationale for a +1.5 °C Limited BECCS scenario.

Answer:

We appreciate your assessment and have tried to clarify why inclusion of this scenario in the main analysis in our view would not strengthen the manuscript.

In ISIMIP2b the climate trajectory of RCP2.6 can only be reached via large scale biomass plantations with partial CCS. The scenario you are suggesting (RCP2.6 climate, but RCP6.0 land use) would thus not be internally consistent.

We did run this inconsistent scenario to distinguish between drivers of water stress difference (see Figure 4 and Methods section “Attribution of drivers for water stress differences”). It can show the pure climate effect on water stress, when being compared with the scenario CC (RCP6.0 climate, and RCP6.0 land use).

However contrary to the CC scenario, the climate trajectory would not be reached without the significant negative emissions from BECCS (i.e. it would not represent a consistent climate-land use scenario). Therefore we did not discuss it at length (for example include it in Figure 1), because we believe that this could make the impression that such a scenario next to the other consistent scenarios would also be feasible, which is not the case.

For the next iteration of the ISIMIP project a scenario like IPCC SR1.5 P1 might be included, including enforcement of the required large scale socio-economic innovations and resulting effects for the land system.

On clarity, I would first like to state that the paper, in terms of language, is well written and understandable. Substantial language edits are unnecessary. When I seek more clarity, it is rather due to the complexity of the models involved. I seek further guidance on a number of issues, starting with feedbacks and uncertainty:

1. How do you deal with the fact that cloud formation and rainfall are two aspects that are highly uncertain in a changing climate? Maybe you’ve addressed this already in the text (and I’ve failed to understand it). If you have, could you please make it more explicit and understandable to a general audience? If you haven’t addressed it, I think it deserves attention. To what extent does it affect how to understand your results? And is there, for example, a difference in certainty of cloud formation and rainfall patterns in a climate with +1.5 °C average surface warming (much closer to today’s approx. +1.0 °C warming) compared to +3.0 °C?

Answer:

Atmospheric processes like cloud formation and rainfall are not modeled by LPJmL but inherent to the GCMs, results of which we use as an input. We think this question cannot be treated fully in an impact analysis like ours (as it is about the GCM structures and parameters, analyzed elsewhere).

The precipitation projections from the GCMs HadGEM2-ES, MIROC5, GFDL-ESM2M and IPSL-CM5A-LR vary to a certain degree (also in simulations for 1.5°C), as can be seen in Figure S9 for many regions. This uncertainty points to the fact that some processes in the atmosphere are treated differently in different models, or at least cannot be resolved at typical resolutions of GCMs. In recognition of these differences, we use a GCM ensemble to consider this uncertainty to a certain degree. This information is now added to the Introduction (lines 106-110).

2. Does your model explicitly account for tipping points? Or is this perhaps implicitly accounted for in the forcing-data for your model?

a. If yes, at what temperature levels are crucial tipping points expected to be triggered? What kinds of bifurcated hysteresis are represented in your scenarios? Does the irreversibility of perturbed new climates, after tipping points, affect your conclusions about water stress in the Limited BECCS +3.0 °C scenario? As a consequence of irreversible climate change, would water stress (understood partially as an impact of climate change) also be irreversible? And could water stress in the Significant BECCS and Significant BECCS+SWM scenarios perhaps be reversible, if the climate forcing is reduced beyond +1.5 °C? Is this important to note?

b. If no, what does this mean for how to understand your results? What does the literature on tipping points have to say about triggering such in a 3.0 °C world? Would +3.0 °C lead to runaway climate change or irreversible system change that makes water stress even worse than indicated by your findings? Would this affect your conclusions about the Significant BECCS with or without SWM scenarios?

3. Does your model/do the model forcing include/represent reversible feedbacks, i.e. other than cloud formation and irreversible feedbacks triggered through crossing tipping points, that would severely affect hydrology?

Answer 2/3:

Tipping points that might be triggered below 3°C of GMT warming are: the West-Antarctic icesheet, the Greenland icesheet, the Arctic summer sea-ice, Alpine glaciers and Coral reefs (Steffen et al., 2018), with the effect that climate change could accelerate and/or become irreversible. Tipping points however are an area of ongoing current research, and they are not well (if at all) represented in GCMs. Therefore one can only associate approximate likelihoods of when they might be triggered and what effects this could have (most of them would also play out over much longer timescales than until 2100).

The analysis of tipping points or more detailed processes in a coupled earth system is relevant but surely elaborate future work, as it requires substantial analyses of other types, involving improvements of GCMs and coupled biosphere-climate models.

Our study focus is different, therefore we confine our analysis to a standard set of GCM projections and the effect that large scale irrigation on biomass plantations would have.

We now explain this in more detail in the Methods section “Climate and land use change scenarios” (lines 300-305).

References:

Steffen, W.; Rockström, J.; Richardson, K.; Lenton, T. M.; Folke, C.; Liverman, D.; Summerhayes, C. P.; Barnosky, A. D.; Cornell, S. E.; Crucifix, M.; Donges, J. F.; Fetzer, I.; Lade, S. J.; Scheffer, M.; Winkelmann, R. & Schellnhuber, H. J.

Trajectories of the Earth System in the Anthropocene

Proceedings of the National Academy of Sciences, **2018**, 201810141

4. A specific form of impact of climate change that I would guess affects water stress quite substantially is sea level rise. Do you account for this? What is the difference of sea level rise in a +1.5 °C world compared to a +3.0 °C world? Loss of land and salt water penetration? While sea level rise is linked to both point 2 and 3 above, I thought I'd mention it separately. I've seen some quite remarkable estimates of loss of land and forced displacement in the wake sea level rise induced by +3 °C warming, but I'm not qualified to assess their validity.

Answer 4:

Sea level rise induced reduction of agricultural area or productivity is not included in this study. We mention this now in the additional paragraph added to the Methods section “Climate and land use change scenarios” (lines 301-302).

5. Are all three scenarios that you explore attached with the same level of likelihood of achieving 1.5 °C and 3.0 °C?

Answer 5:

GCMs were forced by the same RCP, they arrive at different global mean temperatures in 2100. When averaging the 2100 global mean surface temperature changes compared to preindustrial values of the GCMs (Frieler et al. 2017), we arrive at 1.68°C for RCP2.6 and 3.15°C for RCP6.0, which were rounded to the closest half-degree. Associating likelihoods to this small number of projections seems not possible. We added these numbers to the Methods section “Climate and land use change scenarios” (line 306).

The biomass production aims at fulfilling the energy and negative emission demand prescribed in the underlying SSP-RCP scenarios and thus does not add any more uncertainty.

6. I assume that the Significant BECCS and Significant BECCS+SWM scenarios both are assumed to lead to 1.5 °C (with the same level of likelihood), i.e. that they result in roughly the same levels of concentration of greenhouse gases in the atmosphere? This is unclear, i.e. particularly the extent to which the Significant BECCS+SWM scenario is expected to lead to 1.5 °C. It is clear that both lead to a radiative forcing of 2.6 w/m², but it remains unclear to me (as a reader with limited understanding of climate modelling) if this automatically means that they are expected to lead to 1.5 °C or if the forcing of the LPJmL model with input from four global circulation models means that you import different assumptions about climate sensitivity in the different scenarios or somehow use different likelihoods.

Answer 6:

LPJmL is forced with the same climate trajectories in the BECCS and BECCS+SWM case. This is possible, since the bioenergy irrigation level is chosen so that a similar biomass harvest (and thus energy demand and negative emission trajectory) is reached which would result in the same climate trajectory. We have modified the introduction of scenario BECCS+SWM to make this more clear (lines 93-96).

Please review my comments above in context of my limited understanding of climate modeling, whether using GCMs or IAMs. I also note that you, on lines 209-210, introduces a form of disclaimer that the Earth system is much more complex than your model. That your model is not as complex as the Earth system is quite obvious, but I do think that you may want to explicitly address some of the consequences of (the completely necessary) failure to capture all of the Earth system’s complexity for how to understand your findings. I don’t ask of you to quantify responses to all of the issues I raise above, as this would probably not be doable within your model framework. But I do ask you to reflect, in qualitative terms, on their potential significance for how to interpret your results, to the extent that you find the comments relevant.

Answer:

This summarizes your earlier comments (1-6) concerned with our scenario approach. We have added a collective paragraph to the Methods section “Climate and land use change scenarios”, explaining our scenario approach (lines 297-311).

On clarity, again, I also seek further guidance on issues relating to assumptions made regards BECCS:

7. Not all biomass-energy is coupled with CCS in IAM scenarios. Are you actually investigating scenarios of irrigated biomass plantations (regardless if the biomass is used in operations with or without CCS), using irrigated biomass as a proxy for BECCS? The article is ambiguous on this point, but I sometimes feel like biomass is made equal to BECCS. I’m sure you don’t, but it deserves further clarification. This connects to your proposed title for the paper. The title should probably signal the importance of BECCS that you stress

throughout the main body text. In its present form, the title seems slightly off target, unless, of course, what you actually do is to investigate biomass—water stress linkages rather than biomass-limited-to-BECCS—water stress linkages. If it is actually the former, you should make this much more explicit in the paper.

Answer 7:

The underlying IAM scenarios do require biomass harvests for several purposes. Some of it follows the whole BECCS supply chain including CCS and some does not. For our analysis this does not matter, as we focus on the production of the required biomass on agricultural plantations.

We now make this more clear in the Introduction (lines 53-57) and replace “BECCS” with “biomass production” or “biomass plantations” where possible.

8. You discuss the application of BECCS in production of electricity and liquid biofuels. These are promising activities that already emits high levels of biogenic CO₂. But is waste to energy included represented by your model/forcing? Both electricity (from incineration of waste) and liquid biofuels (from agricultural residues) are promising candidates for BECCS and would not, I guess, increase demand for bioenergy (other than, perhaps, to fuel the energy penalty for carbon capture and transport from waste incineration and for transport in bioethanol production). It could complement dedicated biomass boilers and cofired biomass-fossil plants with CCS.

9. Similarly, does your approach account for the BECCS potential in production of gaseous biofuels and in pulp and paper? These activities have a high potential for BECCS, again without BECCS being the main driver behind biomass demand (the demand is for fuel and paper, with CCS being a potential technology to reduce emissions linked to these activities to the extent that they can become negative; the world population, however, would likely not by more paper just to capture and store biogenic CO₂ in order to resolve climate change).

Answer 8 and 9:

By using external land use trajectories, we inherit the external biomass demand from the ReMIND-MAGPIE models, which includes also other feedstock than the harvest of biomass plantations for BECCS. The model version used by ISIMIP2b includes agricultural residues additional to the biomass from dedicated plantations. There might be some additional potential to reduce raw biomass demands by utilization of organic wastes.

The ReMIND model includes fuel production via two pathways of biomass-to-liquid-fuels and biomass-to-hydrogen, which we subsume to bio-fuels in the Introduction. Since these technologies still require biomass input and just replace a (small) portion of the processing chain, this would not change our analysis.

Considering comment 8 and 9, we added a new paragraph to the Methods section “Determining the bioenergy irrigation amount” (lines 347-350).

On the importance of a 1.5 °C with limited BECCS scenario:

10. The above leads me to a second line of thought, which if you agree to it, would require more revisions. In my view, it would, however, also substantially improve your paper. On line 50-51, you seem to assume that BECCS drive demand for biomass. But, to my knowledge, what happens if you exclude BECCS from models’ technology portfolios is that biomass demand increases. This is an effect of a need to phase out fossil fuels earlier than would otherwise be necessary, leading to an increased demand for liquid biofuels to decarbonize the transport sector. For the same temperature goal, say 1.5 °C, BECCS may thus serve to lessen rather than elevate the demand for biomass in IAM scenarios.

Would it not, therefore, be suitable to compare a +1.5 °C Significant BECCS scenario not only to a +3.0 °C Limited BECCS scenario but also to a +1.5 °C Limited BECCS scenario?

If explored in detail, a quite acceptable compromise between fully taking this comment into account and produce a completely new scenario, and to completely neglect this comment,

would be to use previous literature on this topic reflect upon how to interpret your findings in view of the likely result of producing a +1.5 °C Limited BECCS scenario (without actually doing so).

Answer 10:

This is an interesting argument. It would however mean that such a “+1.5 °C Limited BECCS” scenario would have an higher biomass demand than the one we use, leading to potentially even more water stress and thus not changing the main message of the study.

We unfortunately cannot include the suggestions into our analysis, since the changes in available technologies would require a complete rerun of the ReMIND simulations for ISIMIP2b which is not available at the moment.

We can however forward any study supporting Dr. Fridahls argumentation to the ISIMIP modeling team for future projects.

Smaller revision points:

Line 25: Consider clarifying that “will” is conditional to the model design. “Will” seems to be a strong word in context of scenarios, even if preceded by the qualifier “suggest”.

Line 26: Consider rephrasing to improve the clarity of the sentence. You could express this as something along the lines of: “The increased water stress induced by irrigated BECCS may even exceed avoided water stress achieved by using BECCS to limit global warming to 1.5 °C instead of 3.0 °C”.

Line 23-26: In making revisions on line 25 and 26, consider breaking the long sentence on line 23-26 into two sentences to facilitate the reading.

Answer:

We struggled with the Abstract word limit here. By slightly exceeding the word limit, we changed this to:

“By considering a widespread use of irrigated BECCS, global warming by the end of the 21st century could be limited to 1.5°C compared to a climate change scenario with 3°C. However, our results suggest that both the associated global area and population living under severe water stress in the BECCS scenario would double compared to today and even exceed the impact of climate change.”

Table 1, second row: The asterisk is unexplained, lacks a “sibling” that explains what is meant by it.

Answer: We removed the asterisk, as this is explained in the caption.

Line 81: Clarify if you study scenarios or a scenario.

Answer: Replaced by “a scenario”. This might be easier to understand, even though this scenario consists of several simulations in the background.

Line 101-105: Unclear. I think that it would be better to express the relative (percentage) increase reported on line 103 in absolute terms. As you do on line 117, for example. You could then also express this as the share of global population in 2090–99. Or as a percentage increase of the absolute numbers. In the current format, it is hard to understand what you mean when you say that the you’ve included the effect of population increase in the. It only becomes clear when studying the numbers in Figure 1.

Answer: We adopted your suggestion to express the increase in area and population under water stress in absolute terms (lines 122-128). This way also the effect of population change should become clearer.

Figure 1: Consider facilitating the reading of figures when viewed in greyscale. This shouldn’t

be too hard for Figure 1, at least, even if worse (and therefore should probably be ignored) for the rest of the figures.

Answer: We have introduced a darker blue tone (BECCS+SWM) to better distinguish it from the other bars in a monochrome-print.

Line 160: It is hard to make sense of the statement that in inter-model variability, one CGM is in agreement. Do you intend for this to mean that none of the four GCMs agree in those cases? Or do you actually mean to say “...(two or three GCMs agree)...” rather than “...(one or two GCMs agree)...”? In any case, maybe you could make this clearer to the reader.

Answer: We have corrected this to “no or only two GCMs agree” and also changed it in Figure 4.

I really enjoyed reading your paper and hope that some of my comments will be helpful in guiding revisions.

Best, Mathias Fridahl, Linköping University

REVIEWERS' COMMENTS

Reviewer #1 (Remarks to the Author):

thank you for addressing my comments. Looking forward to read the published article.

Reviewer #3 (Remarks to the Author):

Dear Fabian Stenzel,

Thanks for a thorough review. The paper is still rather dense matter to the general reader but many of my question marks have been straightened out and I think that the clarity of the paper has improved greatly. I have no further questions or comments.

Best, Mathias Fridahl